# The UK Biobank mental health enhancement 2022: Methods and results

**Katrina A.S. Davis**[1,2]\*, **Jonathan R.I. Coleman**[1,2], **Mark Adams**[3], **Gerome Breen**[1,2], **Na Cai**[4,5,6], **Helena L. Davies**[1,7,8], **Kelly Davies**[9,10], **Alexandru Dregan**[1], **Thalia C. Eley**[1,2], **Elaine Fox**[11], **Jo Holliday**[9,10], **Christopher Hübel**[1,12,13], **Ann John**[14,15], **Aliyah S. Kassam**[1], **Matthew J. Kempton**[1], **William Lee**[16], **Danyang Li**[1], **Jared Maina**[1], **Rose McCabe**[17], **Andrew M. McIntosh**[3], **Sian Oram**[1], **Marcus Richards**[18], **Megan Skelton**[1], **Fenella Starkey**[9,10], **Abigail R. ter Kuile**[1,18], **Laura M. Thornton**[19], **Rujia Wang**[1], **Zhaoying Yu**[1], **Johan Zvrskovec**[1,2], **Matthew Hotopf**[1,2]

1 King's College London, London, United Kingdom, 2 South London and Maudsley NHS Trust, London, United Kingdom, 3 University of Edinburgh, Edinburgh, United Kingdom, 4 ETH Zurich, Zurich, Switzerland, 5 Helmholtz Munich, Neuherberg, Germany, 6 Technical University of Munich, Munich, Germany, 7 Copenhagen University Hospital – Mental Health Services CPH, Copenhagen, Denmark, 8 Mental Health Center Sct. Hans CPH, Copenhagen, Denmark, 9 University of Oxford, Oxford, United Kingdom, 10 UK Biobank, Stockport, United Kingdom, 11 University of Adelaide, Adelaide, Australia, 12 Aarhus University, Aarhus, Denmark, 13 German Red Cross Clinics, Berlin, Germany, 14 Swansea University, Swansea, United Kingdom, 15 Public Health Wales NHS Trust, Cardiff, United Kingdom, 16 Cornwall Partnership NHS Foundation Trust, Bodmin, United Kingdom, 17 City St George's University of London, London, United Kingdom, 18 University College London, London, United Kingdom, 19 University of North Carolina, Chapel Hill, United States of America

\* katrina.davis@kcl.ac.uk

## Abstract

### Background

This paper introduces the UK Biobank (UKB) second mental health questionnaire (MHQ2), describes its design, the respondents and some notable findings. UKB is a large cohort study with over 500,000 volunteer participants aged 40–69 years when recruited in 2006–2010. It is an important resource of extensive health, genetic and biomarker data. Enhancements to UKB enrich the data available. MHQ2 is an enhancement designed to enable and facilitate research with psychosocial and mental health aspects.

### Methods

UKB sent participants a link to MHQ2 by email in October-November 2022. The MHQ2 was designed by a multi-institutional consortium to build on MHQ1. It characterises lifetime depression further, adds data on panic disorder and eating disorders, repeats 'current' mental health measures and updates information about social circumstances. It includes established measures, such as the PHQ-9 for current depression and CIDI-SF for lifetime panic, as well as bespoke questions. Algorithms and R code were developed to facilitate analysis.

**Data availability statement:** Data cannot be shared publicly for participant confidentiality, but can be accessed subject to conditions by application to UK Biobank https://www.ukbiobank.ac.uk/enable-your-research. Supporting information for reproducing variables is available https://osf.io/5a8yu/

**Funding:** No specific funding was received for this work. KASD was supported by the National Institute of Health and Care Research (NIHR) Maudsley Biomedical Research Centre at South London and Maudsley National Health Service (NHS) Foundation Trust and King's College London (NIHR203318, https://www.maudsleybrc.nihr.ac.uk/about-us.html). The views expressed are those of the authors and not necessarily those of the NIHR or the Department of Health and Social Care. No funder had any role in study design, data collection and analysis, decision to publish, or preparation of the manuscript.

**Competing interests:** AJ received a fee to talk at the centenary of the Scottish Action for Mental Health (SAMH). The other authors declare no conflict of interest. This does not alter our adherence to PLOS ONE policies on sharing data and materials.

**Abbreviations:** AUDIT, Alcohol Use Disorder Identification Tool; CIDI-SF, Composite International Diagnostic Interview Short Form; DSM-IV, Diagnostic and Statistical Manual of Mental Disorders 4th ed.; EQ-5D-5L, EuroQol questionnaire of 5 dimensions and 5 levels, plus VAS; GAD-7, Generalised Anxiety Disorder scale (for current GAD); MHO, Mental Health Outcomes [consortium]; MHQ, Mental health questionnaire; OSF, Open Science Framework; PHQ-9, Patient Health Questionnaire (for current depression); R, Statistical software; UKB, UK Biobank; VAS, Visual Analogue Scale (used in EQ-5D-5L).

## Results

At the time of analysis, MHQ2 results were available for 169,253 UKB participants, of whom 111,275 had also completed the earlier MHQ1. Characteristics of respondents and the whole UKB cohort are compared. The major phenotypes are lifetime: depression (18%); panic disorder (4.0%); a specific eating disorder (2.8%); and bipolar affective disorder I (0.4%). All mental disorders are found less with older age and also seem to be related to selected social factors. In those participants who answered both MHQ1 (2016) and MHQ2 (2022), current mental health measure showed that fewer respondents have harmful alcohol use than in 2016 (relative risk 0.84), but current depression (RR 1.07) and anxiety (RR 0.98) have not fallen, as might have been expected given the relationship with age. We also compare lifetime concepts for test-retest reliability.

## Conclusions

There are some drawbacks to UKB due to its lack of population representativeness, but where the research question does not depend on this, it offers exceptional resources that any researcher can apply to access. This paper has just scratched the surface of the results from MHQ2 and how this can be combined with other tranches of UKB data, but we predict it will enable many future discoveries about mental health and health in general.

## Introduction

Mental health disorders cause considerable suffering for a substantial portion of the population. There is still a lot about mental health and disorder that we do not understand, and there is an urgent need for research to improve our ability to promote good mental health or treat disordered mental health [1,2]. Early mortality of people with severe mental disorders from physical health conditions is likely part of a reciprocal relationship between poor physical and mental health, which reinforces the importance of considering mental health in wider medical research [3–5].

The UK Biobank (UKB) cohort study was established with the scale and depth to help researchers grapple with complex questions about conditions that are life-threatening or disabling in adults [6,7]. Such conditions seldom have a single cause; instead, their onsets and prognoses are associated with many risk factors, including genetic variants at multiple loci, exposures at any time from conception onwards, and interactions between them [8]. Many conditions also tend to cluster, causing further problems in investigating specific causal mechanisms [9]. These features of high complexity are true for most mental disorders [10–12]. Traditional epidemiological designs of modest sample sizes and limited data capture are not well suited to these challenges [13]. The very large sample size and detail of data in UKB offers great potential for investigating many facets of mental disorders in adults.

UKB recruited over half a million individuals aged 40–69 between 2006 and 2010. The baseline assessment was extensive, with physical measures, biological sample collection (blood and urine in all, saliva in 100,000), a touchscreen questionnaire and an interview that gathered information on early life and current behaviour, as detailed on their website [14]. An impressive list of potential biomarkers has been established, and participants' approximate addresses have been used to provide measures such as area deprivation, access to green spaces and air pollution [15]. A subset of participants has been characterised even further through 'enhancements', including via multimodal imaging (including neuroimaging), activity/sleep monitoring, and cognitive testing. Further data, including information on health outcomes, have been obtained through consented linkage to routinely collected information, including hospital in-patient statistics, cancer registry, COVID-19 vaccination data, death certification, and data from the primary care electronic health records for a subset of the cohort [16].

To realise the potential of UKB for mental health research, and mental aspects of physical health, a mental health outcomes (MHO) consortium was created, which made recommendations for an online questionnaire about current and past mental disorders and associated features. In 2016–17 participants were invited via email and postal newsletter to complete this first web-based questionnaire (MHQ1), leading to responses from over 157,000 individuals by July 2017 [17]. The core of the questionnaire was an assessment of lifetime depression and generalised anxiety status, which allowed genome-wide association studies (GWAS) for these disorders [18,19], but also facilitated non-genetic studies, touching on many other aspects of mental health research, as detailed in a recent review of the use of UKB for mental health research [20].

Recognising the changeability of mental health and the limitations of a single questionnaire to capture complex diagnoses, the MHO consortium proposed a second mental health questionnaire (MHQ2). Alterations were made to reflect the changing priorities of research following the COVID-19 pandemic. Like its predecessor, MHQ2 was not a distinct entity, but a collection of 'modules' that contained measures on a particular diagnostic or exposure topic. When choosing measures, consideration was given to comparability and complementarity with other cohort studies in the UK [21] and elsewhere [22], and experiences of using MHQ1 data. Scales on current depression, anxiety and alcohol use were repeated from MHQ1 to allow examination of possible fluctuation of these symptoms. MHQ2 also aimed to:

- Enrich the lifetime depression phenotype including treatment response
- Investigate new disorders: panic disorder and eating disorders
- Update aspects of participants' social circumstances.

## Materials and methods

### Aim

This paper outlines the content of the MHQ2 questionnaire, describes the cohort, and gives an overview of participant responses. We aim to report (A) who responded to the MHQ1 and MHQ2, (B) the mental health phenotypes and social factors ascertained in MHQ2, (C) changes in the current mental health of the cohort, and (D) the consistency of lifetime phenotypes.

### UK Biobank

UKB invited people aged 40–69 years living close to one of 22 assessment centres in England, Scotland, and Wales between 2006–2010 to take part. 503,309 (5.5%) agreed, and 91% remain in the cohort. Three main sources of data in UKB are: baseline assessment and samples; linkages such as hospital inpatient records; and 'enhancements' such as imaging and the MHQs.

Linkage and enhancements have variable coverage and have been conducted at various timepoints, so caution is needed in the analysis and interpretation of such data, as shown in a recent timeline [20]. Participants in UKB differ from the wider UK population by more likely being female, having higher educational attainment, and living in areas with less deprivation, though more urban than average [23]. The ethnicity of participants is predominantly White British, with small numbers of participants from other ethnicities [24]. The health and health behaviours of the UKB participants are better than average, with less smoking, lower incidence of cancer and lower mortality [23,25].

UKB operates inside the ethical framework of the UK Health Research Authority. Participants gave consent for their data to be used and can withdraw at any time, with an option to withdraw their existing data from future analysis. Research Ethics Committee opinion has been sought for UKB and each enhancement from the North West - Haydock Research Ethics Committee, including November 2022 (21/NW/0157, amendment 07).

UKB data can be accessed by submitting a research protocol for approval [14], subject to UKB procedures including that researchers have no ability to identify participants. Over 4,000 research proposals had been approved by January 2024. Most of the data in this paper were accessed through UKB-approved research number 82087, downloaded 23/10/2023, with withdrawals downloaded 11/06/2024. The exception was for the participation numbers, which were provided by UKB's scientific team with a censoring date of 11/06/2024.

## Questionnaire administration

UKB participants who had a valid email address on record were sent an invitation email in October to November 2022, which included a hyperlink to a personalised questionnaire. If there was no response, a reminder email was sent two weeks after the initial invitation. Respondents who started, but did not complete, the questionnaire were sent a reminder two weeks after they last accessed it. Finally, a second invitation was sent to non-responders two months after their previous one (around January 2023). All participants were also able to access the online questionnaire via the UKB participant website, as explained in annual postal newsletters (sent to people with no email address). Respondents were asked to enter their date of birth as an identity check, and the data for any respondents whose entered date of birth did not match the information previously provided are not released. We define someone as having completed the MHQ2 if they completed the mental health modules shown in Table 1 (there was also a COVID-19 module) and their MHQ2 data was available, which excludes any participants who failed the identity check or withdrew their data between answering the questionnaire and our data access. This means that 'non-completed' includes a few participants who have answered some or all of the modules.

## Questionnaire development

The questionnaire domains were agreed upon by the consortium after a prioritisation survey and feedback from the Psychiatric Genomics Consortium (PGC) leads [26]. Measures were discussed internally and with investigators who had experience with several other studies, including Genetic Links to Anxiety and Depression (GLAD) [27], The Scottish Health Research Register & Biobank (SHARE) [28], English Longitudinal Study of Ageing (ELSA) [29], the Eating Disorders Genetics Initiative (EDGI) [30], and the Adult Psychiatric Morbidity Survey (APMS, or Survey of Mental Health and Wellbeing) [31].

The measures chosen are provided in Fig 1, which shows how some modules were repeated exactly from MHQ1, and some were changed. For example, MHQ1 asks participants if they had been diagnosed with "panic attacks or panic disorder", but in MHQ2, people are separately asked whether they have been diagnosed with "panic attacks" and then "panic disorder" to discriminate between them. Such changes mean some sections were not exactly equivalent, so were labelled 'repeated with changes' in Fig 1. To maximise the acceptability of the questionnaire length, some of the measures from MHQ1 were not repeated. These choices were made according to the relative support in the consortium for inclusion. The

**Table 1. Modules of the UK Biobank MHQ2.**

| Module | Domain | Source/tool | Selected phenotypes defined<br>N = new, R = repeated, R' = repeated with change |
|---|---|---|---|
| *Mental Health* | Screening | MHO consortium | Any self-report (SR) diagnosis [R'] |
| | Family history | MHO consortium, based on other studies | Known family history of mental health conditions [N] |
| *Depression* | Current depression | Patient Health Questionnaire 9-item version (PHQ-9) [34] | PHQ-9 total score (symptoms) [R]<br>PHQ-9 derived depression [R]<br>• Algorithm-based<br>Current depression case [R]<br>• Requires case on both PHQ9 (current) and CIDI-SF (lifetime) depression algorithms |
| | Lifetime depression | Composite International Diagnostic Interview – Short Form (CIDI-SF) [35] depression module, lifetime version [36] PLUS subtype questions | Depression ever [R]<br>Depression with melancholic features [N]<br>Depression with atypical features [N] |
| | Antidepressant and therapy response | MHO consortium, based on other studies | Medication helped [N]<br>Non-medication therapy helped [N] |
| *Mood change* | Lifetime manic symptoms | UKB baseline questionnaire (based on CIDI questions mapping to DSM-IV) [37] | Hypomania/mania ever (symptoms) [R]<br>Bipolar affective disorder type I [R] |
| *Anxiety and panic* | Current anxiety disorder | Generalised Anxiety Disorder Questionnaire - 7 item (GAD-7) [34] | GAD-7 total score (symptoms) [R]<br>GAD-7 derived anxiety disorder [R']<br>• As CIDI-SF for GAD not repeated, not able to require positive for lifetime GAD |
| | Lifetime anxiety disorder (panic) | CIDI-SF [35] panic disorder, lifetime version | Panic attack ever (symptoms) [N]<br>Panic disorder ever [N] |
| *Adverse life events* | Adverse events in childhood | Childhood Trauma Screener – 5 item (CTS-5) [38] | Childhood adverse events [R] |
| | Adverse events in adulthood | MHO consortium, based on APMS and National Crime Survey [39] | Adult abuse events [N]<br>Adverse events 12 months [R'] |
| *Alcohol use* | Alcohol use disorder | Alcohol Use Disorders Identification Test (AUDIT) [40] | AUDIT total score [R]<br>Harmful drinking (12 month) [R] |
| *Cannabis use* | Cannabis use | MHO consortium | Cannabis use ever [R]<br>Daily cannabis use ever [R] |
| *Harm behaviours* | Self-harm and suicidal thoughts | MHO consortium and service-user group | Harm to self ever [R]<br>Suicide attempt ever [R] |
| *Eating patterns* | Eating disorders | MHO consortium, based on other studies | Anorexia nervosa [N]<br>Bulimia nervosa [N]<br>Binge-eating disorder [N]<br>Purging disorder [N] |
| *General health* | General health and functioning | EuroQoL 5 dimensions 5 levels (EQ-5D-5L) plus visual-analogue scale (VAS) [41] | VAS score [R] |
| *Social situation* | Social situation | UKB baseline questionnaire. Includes cohabitation, social contact, employment status | Social isolation [R]<br>Virtually connected [N] |
| | Loneliness | Abbreviated UCLA loneliness scale [42] | Short scale UCLA Loneliness total score [N] |
| *General wellbeing* | Resilience | Brief Resilience Scale [43] | Brief Resilience total score [N] |
| | Subjective wellbeing | MHO consortium | QoL total score [R] |

Algorithms to define 'phenotypes' are available on OSF [33].

| Domain | Not repeated in MHQ2 | Repeated exactly from MHQ1 | Repeated from MHQ1 with changes | Repeated from other questionnaires | New to MHQ2 |
|---|---|---|---|---|---|
| **Depression** | - | Current depression: MHQ-9<br><br>Lifetime depression: CIDI-SF<br><br>Manic symptoms | - | - | Depression subtypes<br><br>Depression treatment |
| **Anxiety and Stress-Related** | Lifetime generalised anxiety disorder: CIDI-SF<br><br>PTSD: PCL-S | Current anxiety: GAD-7 | - | - | Lifetime panic disorder: CIDI-SF |
| **Substance Use and Eating Disorders** | Addictions | Alcohol use: AUDIT<br><br>Cannabis use | - | - | Eating disorders |
| **Transdiagnostic** | Unusual experiences | - | Self-reported clinician diagnosis<br><br>Self-harm and suicidal thoughts | Quality of life | Family history |
| **Reported Events** | - | Adverse events in childhood: CTS-5 | Adverse events in adult life | Social situation | Brief resilience scale<br><br>Loneliness: abbreviated UCLA |

← ————————— MHQ1 ————————— →

← —————————————— MHQ2 —————————————— →

**Fig 1. Summary of the topics included in the UK Biobank MHQ1 and MHQ2, illustrating the overlap.**

modules are also shown in Table 1, which shows the measures used in each domain and the phenotypes that could be derived from these measures. A more detailed explanation is available on the UKB data showcase [32].

### Algorithms and code in R

Symptoms, questionnaire scores, experiences and (probable) diagnostic status that can be derived from the questionnaire are generically termed 'phenotypes' (with no assumption of genetic causation). For each module or group of modules, small groups of MHO members with expertise in those subjects produced algorithms to derive these phenotypes from questionnaire responses. These were then checked and edited to create a consistent style and circulated to the whole consortium for comments. The final document is available on our Open Science Framework (OSF) page for this project [33].

A team of coders, including members of the MHO and junior colleagues, worked in small groups to render the algorithms for each module in R code, with a check by another coder outside of that module subgroup. Lastly, an over-arching script was created that brought the code from all the modules' subgroups together, including a final few definitions that required responses from multiple modules. The algorithms and code for processing MHQ1 and MHQ2 are available on our OSF and GitHub pages [33,44].

A clinical diagnosis is informed by diagnostic criteria, including the lack of alternative explanations and the presence of impact on the patient. However, when we use questionnaires, we are unable to take wider issues into account. The criteria we use for the "lifetime depression" phenotype, for example, are therefore referred to as quasi-diagnostic because they are informed by diagnostic criteria, but fail to fully examine a participant. When we refer to a group with "lifetime depression", we mean only that they met the criteria for lifetime depression on our measure, not that they were fully assessed.

## Analysis

We planned a descriptive summary of the questionnaire respondents and responses. We refer to the proportion of cases in the respondents. Due to the issues with representativeness, we do not attempt to make inferences about population prevalence, so hence do not include confidence intervals. Significance testing on large cohorts may identify many small differences between comparators that have very little effect [45], therefore we have emphasised the magnitude of difference rather than calculating p-values. Percentages are given in whole percentage points unless under 10% when one decimal place is used, as recommended for clarity [46]. Where rare disorders and detailed breakdown of age or ethnicity intersected to make a cell size of less than 10 participants we have merged fields out of caution for confidentiality [47].

Our analysis according to our aims were:

A: Describing Respondents

We describe participants in the 2022 MHQ2 wave, including respondents of MHQ2 regardless of whether they participated in MHQ1. The 2016 MHQ1 wave and the full original UKB cohort are presented for comparison. The age variable was handled differently for the three groups. Age for the MHQ2 wave was their age on completion of the MHQ2, and for the MHQ1 wave age at completion of MHQ1. For the UKB cohort as a whole, age represents the age they would have been at the median date of MHQ2 completion, including those who died before this date, excluding those with missing baseline data (n = 2).

B: Mental Health Phenotypes and Social Factors

We present numbers with lifetime mental health disorders in MHQ2 and describe the personal characteristics in the groups responding to different mental health disorders. Selected social data - social isolation, loneliness, resilience and health-related quality of life – is reported, stratified by lifetime mental disorder phenotypes.

C: Changes in Current Mental Health

We describe and visualise the proportion of respondents who were positive for specified current mental disorders in 2022 and in 2016 in those participants who answered in both waves. Participants are defined as having current mental disorder if they meet case criteria according to PHQ-9 (depression), GAD-7 (anxiety) and AUDIT (harmful alcohol use). These are analysed in age-group and sex-stratified blacks, and plotted as line graphs for each disorder, showing the proportion positive by age-group for each sex and each wave. Age groups in seven-year blocks, which reflects the amount of time between the two waves, such that most participants move one age block between MHQ1 and MHQ2, although not exact. Relative risk for 2022 compared to 2016 was calculated by using 'proportion positive' as the risk of disorder in each cohort and then risk in 2022 by that in 2016. This is done in age-and-sex stratified groups such that those aged, for example, 73–79 years in 2016 are compared with those aged 73–79 in 2022 for the four age groups where possible and also for the cohort overall.

D: Consistency of lifetime phenotypes

We compare 'lifetime status' in 2022 compared to 2016 in those who completed both waves, to give test-retest stability. Status for each participant was meeting case criteria for depression ever (CIDI-SF lifetime), bipolar ever (adapted CIDI), self-harm ever (self-report), cannabis ever (self-report) and any clinician diagnosis (self-report). Percentage agreement for each status was calculated as status agreeing in 2016 and 2022 (both positive or both negative) as a proportion of the cohort, and Cohen's kappa for each status, with kappa being agreement corrected for agreement by chance [48]. Where a respondent was negative in 2016 but positive in 2022, we also took into account the date that the respondent gave for onset in MHQ2 (except for clinician diagnosis, where this was not asked). This helps separate

these respondents into those whose status could have been picked up in the MHQ1 wave (date of onset from MHQ2 was before 2016), from those who might not have done (either date of onset after start of 2016 or answered 'don't know' to date of onset). We decided to include unknown with the later onsets, despite no information to suggest these are more likely to be late onset, to explain the greatest amount of test-retest variability.

## Results

### Sample

Data provided by UKB's scientific team showed that, after accounting for deaths and withdrawals, 457,653 participants were eligible to complete the MHQ2 (Fig 2). Of these, UKB was able to invite 329,902 (72%) participants by email, and 53% of those contacted answered at least one module of the questionnaire, alongside a small number (n = 530) who accessed the questionnaire through their participant website. 175,266 participants completed at least one module with 169,252 completing all the mental health modules in the questionnaire (97% of those who started). The response dates cluster around early November 2022 (approx. 80%), when most emails went out, with a smaller peak in approximately January 2023 when reminder emails were sent.

Fig 3 shows that the 169,252 UKB participants who completed MHQ2 consisted of 111,272 who had also completed MHQ1 plus 57,980 who had not. This gives a sample of 111,272 who have longitudinal data (both waves), and a sample of 215,250 who have data for either MHQ.

### A: Describing respondents

Data available to researchers was of 169,253 respondents. These numbers vary slightly to the data from the UKB scientific team (169,252), and may vary due to late completion, quarantine of questionnaires when the identity check was failed (and resolution thereof), and the small delay in communicating participants withdrawing from UKB. Future researchers may find different numbers, also due to these factors.

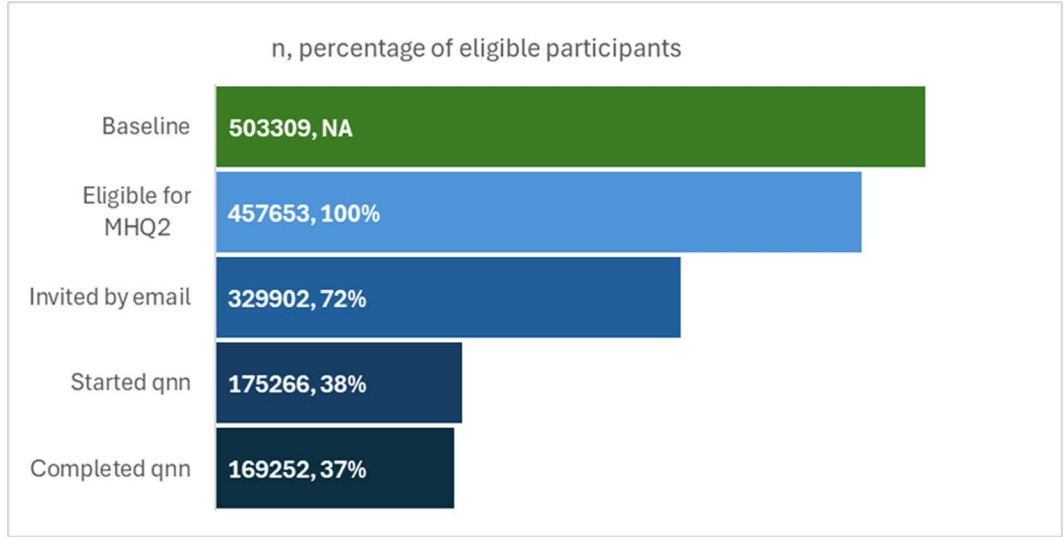

**Fig 2. Number of UKB participants invited, starting and completing the MHQ2 questionnaire, as a proportion of those eligible for the questionnaire. UKB scientific team numbers (n = 165,262).** Participants could participate in questionnaire without being invited. Started qnn = all respondents who completed at least one module on MHQ2; Completed qnn = respondents who completed all mental health modules in MHQ2.

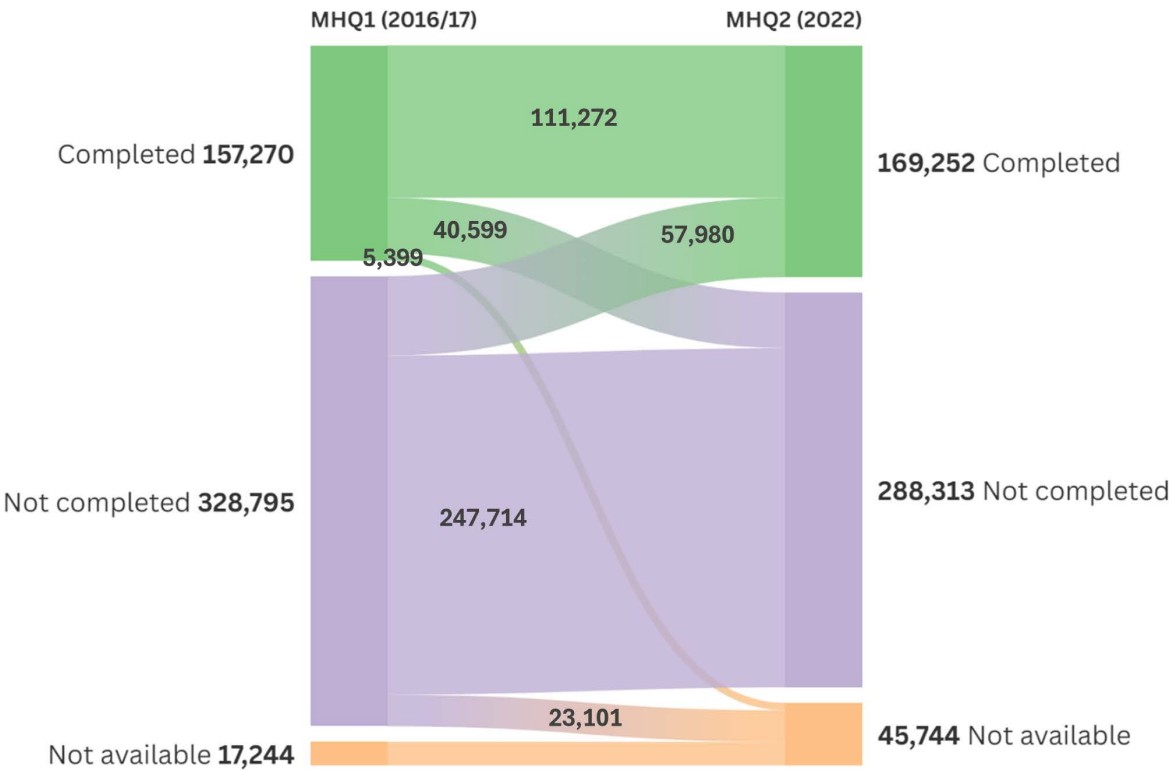

**Fig 3. The flow of participants illustrating those completing the first and second mental health questionnaires, the cross-over and participants who have died or withdrawn. UKB scientific team numbers.** Completed MHQ2 = completed all 12 mental health modules; Not completed MHQ2 = available but didn't start MHQ2, started but didn't finish, or failed their identity check; Not available = died or withdrew from the cohort prior to MHQ1/ MHQ2, or withdrew with no further access between MHQ2 and data access.

The characteristics of the respondents in the MHQ2 wave are shown in Table 2. Compared to those for whom we have data from MHQ1, they are an average of five years older (after an interval between MHQ1 and 2 of approximately six years), and very similar in sex, ethnicity, and the distribution of selected social factors. The comparison of the MHQ2 wave with the UKB cohort suggested possible differences from the base cohort. These include respondents being 4.6 percentage points less likely to live in a deprived area, 5.6 percentage points less likely to have a longstanding illness and 2.8 percentage points more likely to be White, rather than a minority ethnic group, and shows that MHQ1/2 may under-represent people from Black and Asian ethnicities.

### B(1): Mental health phenotypes

In Table 3 and the UPSET plot S1 Fig in S1 File, participants are categorised by lifetime disorder status based on symptom-based definitions for depression, panic disorder, eating disorders and bipolar type I. Participants were most likely to be positive for lifetime depression (18%), followed by panic disorder (4.0%), any specific eating disorder (2.8%), and bipolar affective disorder type I (0.4%). Seventy-one percent of the MHQ2 respondents did not meet the criteria for any of the studied lifetime disorders.

The symptom-based definition of panic disorder was included for the first time in the MHQ2 wave. Self-reported clinician diagnosis of panic disorder appears to be much lower (0.2%) than the symptom-based definition (4.0%), although 4.5% reported a clinician diagnosis of a panic *attack*. For context, in the MHQ1, the symptom-based definition of lifetime generalised anxiety disorder (GAD) was met by 7%.

**Table 2. Characteristics of the respondents to each of the mental health questionnaire waves with comparison to the UK Biobank cohort. Research team numbers n = 165,253.**

| Characteristic | MHQ2 wave (2022) | % | MHQ1 wave (2016) | % | UKB full cohort | % | Absolute Difference MHQ2 vs MHQ1 | MHQ2 vs UKB |
|---|---|---|---|---|---|---|---|---|
|  | 169,253 | % | 157,274 | % | 502,177 | % | MHQ2 vs MHQ1 | MHQ2 vs UKB |
| Demographics from baseline | | | | | | | | |
| **Age[1]** | | | | | | | | |
| 45–54 | 1,577 | 1% | 23,449 | 15% | 5,254 | 1% | −14% | −0.1% |
| 55–64 | 44,319 | 26% | 51,830 | 33% | 122,298 | 24% | −6.8% | +1.8% |
| 65–74 | 70,462 | 42% | 70,122 | 45% | 174,121 | 35% | −3.0% | +7.0% |
| 75–84 | 52,572 | 31% | 11,873 | 8% | 198,269 | 39% | +24% | −8.4% |
| >84 | 323 | 0% | 0 | 0% | 2,235 | 0% | +0.2% | −0.3% |
| Median | 70 | | 65 | | 72 | | +5y | −2y |
| **Sex** | | | | | | | | |
| Female | 97,646 | 58% | 89,048 | 57% | 273,184 | 54% | +1.1% | +3.3% |
| Male | 71,607 | 42% | 68,226 | 43% | 228,993 | 46% | −1.1% | −3.3% |
| **Ethnicity** | | | | | | | | |
| White | 163,795 | 96.8% | 152,063 | 96.7% | 471,837 | 94.0% | +0.1% | +2.8% |
| Black | 1,123 | 0.7% | 1,146 | 0.7% | 8,022 | 1.6% | −0.1% | −0.9% |
| Asian | 1,448 | 0.9% | 1,336 | 0.8% | 9,829 | 2.0% | 0.0% | −1.1% |
| Chinese | 375 | 0.2% | 364 | 0.2% | 1,573 | 0.3% | 0.0% | −0.1% |
| Mixed | 863 | 0.5% | 819 | 0.5% | 2,902 | 0.6% | 0.0% | −0.1% |
| Other | 902 | 0.5% | 876 | 0.6% | 4,553 | 0.9% | 0.0% | −0.4% |
| Missing | 747 | 0.4% | 670 | 0.4% | 3,461 | 0.7% | 0.0% | −0.2% |
| Social factors from baseline | | | | | | | | |
| Relative deprivation[2] | 20,015 | 12% | 19,281 | 12% | 82,263 | 16% | −0.4% | −4.6% |
| Degree educated | 75,311 | 44% | 70,940 | 45% | 161,019 | 32% | −0.6% | +12% |
| Rents home[3] | 7,909 | 5% | 7,939 | 5% | 46,403 | 9% | −0.4% | -4.6% |
| Self-reports illness > 1y | 44,296 | 26% | 43,420 | 28% | 159,800 | 32% | −1.4% | −5.6% |
| Smoker | 11,861 | 7% | 11,330 | 7% | 52,940 | 11% | −0.2% | −3.5% |
| Physically active[4] | 62,371 | 37% | 57,788 | 37% | 169,109 | 34% | +0.1% | +3.2% |
| Neuroticism score, mean (sd)[5] | 3.87 | (3.16) | 3.87 | (3.17) | 4.12 | (3.27) | 0.0pt | −0.25pt |

(1) Age in the MHQ2 wave is at date of completion of MHQ2. Age in the MHQ1 wave is at approximate date of completion of MHQ1. Age for full cohort is the age a participant would have been on the median date of MHQ2 completion.

(2) Townsend Deprivation Score of residence ≥ +2 (where 0 is average, and higher scores are more deprived, [23]).

(3) Private or social rented accommodation at baseline.

(4) Reported at least moderate activity for 20 minutes three times per week.

(5) Eysenck neuroticism scale.

Within the eating disorders category were four specific disorders, and respondents could fall into more than one in their lifetime. Anorexia nervosa criteria were met by 1.7% of respondents, purging disorder by 0.9%, bulimia nervosa by 0.4%, and binge-eating disorder by 0.2%. This can be compared to self-report of an eating disorder diagnosis, for example, self-report of anorexia nervosa was 0.5%.

The proportion with any lifetime disorder (symptom-based definition) may be patterned by age, ethnicity and sex. Lifetime disorder appears to be higher in younger age (45–64: 34%), higher in those of Mixed ethnicity at 30%, than White

(21%) or Asian (15%); higher in female (27%) than male (14%). We particularly saw apparent higher rates in females was particularly marked for eating disorders (F: 4.6%, M: 0.4%).

S1 Fig in S1 File shows the degree of comorbidity between the phenotypes in Table 3 with the addition of self-harm ever. The lifetime self-harm phenotype includes 7765 (4.6%) participants, most (57%) also meeting the criteria for another phenotype, most commonly depression. Of those with lifetime depression, 29% also have one or more other phenotype, most commonly self-harm.

## B(2): Social factors

The distributions of social factors are shown in Table 3, with the results in female participants in S1 Table in S1 File and male participants in S2 Table in S1 File. The variables assessed at baseline were area-level deprivation, education ('degree educated'), housing tenure ('rents home'), longstanding illness, smoking, physical activity, and neuroticism – of which area-level deprivation, renting a home and smoking seem likely to be related to lifetime disorder. The group of respondents that met the criteria for eating disorders and panic disorder appear to have similar characteristics to those respondents with depression and bipolar, although it was remarkable that over half of those with lifetime eating disorders had a degree qualification (53%).

Adverse events in childhood were reported by 42% and adverse events in their adult life by 22%. In the group that met no symptom-based definition in the MHQ2, 38% reported a childhood event, increasing to 58% in those with lifetime depression, and rising further for rarer disorders – with the same pattern for events in adult life. The pattern of adverse childhood event report stratified by major phenotypes is the same in both men and women, S1 and S2 Table in S1 File.

Resilience was 'low' on the brief resilience scale in 10% of the group that met no symptom-based definition in the MHQ2 but 34% to 56% in those who met at least one symptom-based definition for a lifetime disorder. There also appeared to be a pattern for higher social isolation and loneliness in those who met at least one symptom-based definitions, but this was less marked than the pattern for low resilience.

Self-rated health was taken from the EQ-5D-5L visual analogue scale, where respondents rate their health 'today' using a slider from 0 to 100 [41]. This rating was an average of 88 for people with no lifetime criteria, 80 for those with depression and eating disorder history, 75 for those with panic disorder and 70 for those with bipolar disorder. More than a quarter of those with lifetime bipolar disorder had marked their health as below 50 on the scale.

## C: Changes in current mental health

Three measures of current mental health were included in both questionnaires: AUDIT for harmful alcohol use "in the last year", PHQ-9 for depression "in the last two weeks", GAD-7 for generalised anxiety "in the last two weeks". These were identical in MHQ1 and MHQ2. Restricting to people with data from both the MHQ1 and MHQ2 waves (n = 111,275), S3 Table in S1 File shows the proportions meeting the criteria, stratified by age and sex at time of answering. This is illustrated in Fig 4–6 where the lines show the proportion of the cohort positive for each disorder across the age range. The figures show that, within each wave, the slope of proportion with depression, generalised anxiety and harmful alcohol use by age is downwards, suggesting current mental disorder is less common with increasing age. Between questionnaires, harmful alcohol use decreased along lines predicted for the age at the later wave (Fig 4). For depression (Fig 5) and generalised anxiety (Fig 6), the curves for the older MHQ2 wave do not follow the age curve to the same extent as for harmful alcohol use, but shift towards greater depression and anxiety in 2022 than the same age respondents in 2016.

S4 Table in S1 File quantifies this a little more by reporting the relative risk of each current mental disorder in 2022 versus 2016 in each sex and age-group stratification and overall. A relative risk of 1 would suggest no change of risk in the group for the current disorder in 2022 compared to 2016 wave, with values above 1 suggesting a higher risk in 2022. The relative risk for every sex and age-group stratified group was above 1, between 1.07 and 2.00, with the latter representing a doubling of the presence of current depression in women aged 73–79 in 2022 compared to 2016. Despite age-group

**Table 3. Respondents categorised by symptom-based criteria, which are not exclusive, with characteristics based on baseline questionnaire (BL) and second mental health questionnaire (MHQ2/Q2). Research team numbers n = 165,253.**

| Characteristic | Overall | | No lifetime criteria | | | Depression | | | Panic Disorder | | | Any eating disorder | | | Bipolar type I | | |
|---|---|---|---|---|---|---|---|---|---|---|---|---|---|---|---|---|---|
| | N | % item | N. | % item | % no dx | N. | % item | % dep'n | N. | % item | % panic dx | N. | % item | % eating dx | N. | % item | % bp I |
| N | **169,253** | 100% | **132,975** | NA | 79% | **31,243** | NA | 18% | **6,703** | NA | 4.0% | **4,762** | NA | 2.8% | **721** | NA | 0.4% |
| **Age[1]** | – | – | – | – | | – | – | | – | – | | – | – | | – | – | |
| 45-54 | 1,577 | 0.9% | 1,041 | 0.8% | 66% | 444 | 1.4% | 28% | 133 | 2.0% | 8.4% | 99 | 2.1% | 6.3% | 15 | 2.1% | 1.0% |
| 55-64 | 44,319 | 26% | 30,781 | 23% | 69% | 11,488 | 37% | 26% | 2,880 | 43% | 6.5% | 2,238 | 47% | 5.0% | 337 | 47% | 0.8% |
| 65-74 | 70,462 | 42% | 55,443 | 42% | 79% | 12,962 | 41% | 18% | 2,642 | 39% | 3.7% | 1,899 | 40% | 2.7% | 264 | 37% | 0.4% |
| 75+ | 52,895 | 31% | 45,710 | 34% | 86% | 6,349 | 20% | 12% | 1,048 | 16% | 2.0% | 526 | 11% | 1.0% | 105 | 0 | 0.2% |
| Median | 70 | – | 71 | – | | 67 | – | | 66 | – | | 65 | – | | 65 | – | |
| **Sex** | – | – | – | – | | – | – | | – | – | | – | – | | – | – | |
| Female | 97,646 | 58% | 71,372 | 54% | 73% | 22,169 | 71% | 23% | 4,913 | 73% | 5.0% | 4,447 | 93% | 4.6% | 473 | 66% | 0.5% |
| Male | 71,607 | 42% | 61,603 | 46% | 86% | 9,074 | 29% | 13% | 1,790 | 27% | 2.5% | 315 | 7% | 0.4% | 248 | 34% | 0.3% |
| **Ethnicity** | – | – | – | – | | – | – | | – | – | | – | – | | – | – | |
| White | 163,795 | 97% | 128,605 | 97% | 79% | 30,306 | 97% | 19% | 6,493 | 97% | 4.0% | 4,598 | 97% | 2.8% | 682 | 95% | 0.4% |
| Black | 1,123 | 0.7% | 921 | 0.7% | 82% | 176 | 0.6% | 16% | 43 | 0.6% | 3.8% | NA | NA | NA | NA | NA | NA |
| Asian | 1,448 | 0.9% | 1,232 | 0.9% | 85% | 184 | 0.6% | 13% | 38 | 0.6% | 2.6% | NA | NA | NA | NA | NA | NA |
| Chinese | 375 | 0.2% | 319 | 0.2% | 85% | 49 | 0.2% | 13% | 11 | 0.2% | 2.9% | NA | NA | NA | NA | NA | NA |
| Mixed | 863 | 0.5% | 603 | 0.5% | 70% | 223 | 0.7% | 26% | 56 | 0.8% | 6.5% | NA | NA | NA | NA | NA | NA |
| Other | 902 | 0.5% | 713 | 0.5% | 79% | 164 | 0.5% | 18% | 36 | 0.5% | 4.0% | NA | NA | NA | NA | NA | NA |
| Missing | 747 | 0.4% | 582 | 0.4% | 78% | 141 | 0.5% | 19% | 26 | 0.4% | 3.5% | NA | NA | NA | NA | NA | NA |
| [Combined][2] | NA | NA | NA | NA | NA | NA | NA | NA | NA | NA | NA | 164 | 3.4% | 3.0% | 39 | 5.4% | 0.7% |
| | | | | | | | | | | | | | | | | | |
| Degree educated (BL) | 75,311 | 44% | 58,565 | 44% | 78% | 14,280 | 46% | 19% | 2,818 | 42% | 3.7% | 2,522 | 53% | 3.3% | 340 | 47% | 0.5% |
| Resides in deprived area (BL)[3] | 20,015 | 12% | 14,659 | 11% | 73% | 4,629 | 15% | 23% | 1,156 | 17% | 5.8% | 848 | 18% | 4.2% | 165 | 23% | 0.8% |
| Rents home (BL) | 7,909 | 4.7% | 5,249 | 3.9% | 66% | 2,365 | 7.6% | 30% | 641 | 10% | 8.1% | 394 | 8.30% | 5.0% | 110 | 15% | 1.4% |
| Self-report illness for>1y (BL) | 44,296 | 26% | 32,101 | 24% | 72% | 10,847 | 35% | 24% | 2,513 | 37% | 5.7% | 1,392 | 29% | 3.1% | 394 | 55% | 0.9% |
| Smoker (BL) | 11,861 | 7.0% | 8,396 | 6.3% | 71% | 3,024 | 10% | 25% | 751 | 11% | 6.3% | 464 | 10% | 3.9% | 116 | 16% | 1.0% |
| Physically active (BL)[4] | 62,371 | 37% | 48,899 | 37% | 78% | 11,741 | 38% | 19% | 2,513 | 37% | 4.0% | 1,538 | 32% | 2.5% | 263 | 36% | 0.4% |
| Childhood adverse experience (Q2)[5] | 71,049 | 42% | 50,170 | 38% | 71% | 18,067 | 58% | 25% | 4,280 | 64% | 6.0% | 3,019 | 63% | 4.2% | 536 | 74% | 0.8% |
| Adult adverse experience (ever) (Q2) | 38,027 | 22% | 23,584 | 18% | 62% | 12,465 | 40% | 33% | 3,124 | 47% | 8.2% | 2,425 | 51% | 6.4% | 399 | 55% | 1.0% |
| Low resilience (Q2)[6] | 26,367 | 16% | 13,103 | 10% | 50% | 11,944 | 38% | 45% | 3,127 | 47% | 12% | 1,640 | 34% | 6.2% | 406 | 56% | 1.5% |
| Social isolation (Q2)[7] | 17,365 | 10% | 12,099 | 9.1% | 70% | 4,675 | 15% | 27% | 1,077 | 16% | 6.2% | 706 | 15% | 4.1% | 119 | 17% | 0.7% |
| Loneliness score, median (IQR) (Q2)[8] | 3 | 3–5 | 3 | 3–4 | | 4 | 3–6 | | 5 | 3–6 | | 5 | 3–6 | | 5 | 3–7 | |

*(Continued)*

**Table 3.** (Continued)

| Characteristic | Overall | | No lifetime criteria | | | Depression | | | Panic Disorder | | | Any eating disorder | | | Bipolar type I | | |
|---|---|---|---|---|---|---|---|---|---|---|---|---|---|---|---|---|---|
| | N | % item | N. | % item | % no dx | N. | % item | % dep'n | N. | % item | % panic dx | N. | % item | % eating dx | N. | % item | % bp I |
| Self-rated health median (IQR) (Q2)[9] | 83 | 72–90 | 85 | 75–91 | | 79 | 65–89 | | 75 | 60–85 | | 80 | 65–90 | | 70 | 49–83 | |
| Neuroticism score, mean (sd) (BL)[10] | 4 | -3.16 | 3 | -2.93 | | 6 | -3.34 | | 7 | -3.36 | | 6 | -3.35 | | 7.03 | -3.42 | |

BL = based on answers to baseline questions; Q2 = based on answers to MHQ2 questions; dx = disorder.

(1) Age at MHQ2 is at date of completion of MHQ2.

(2) To preserve participant privacy, non-White and missing ethnicities are collapsed to a single value ("Combined") if one cell size is smaller than 10.

(3) Townsend Deprivation Score of residence ≥ +2 (where 0 is average, and higher scores are more deprived, [23]).

(4) Reported at least moderate activity for 20 minutes three times per week.

(5) Any domain positive on the Childhood Trauma Screener.

(6) Brief Resilience Scale, low = 1 to 2.99, where scale is 1–5.

(7) Scores on at least two of three isolation questions.

(8) Loneliness score from 3 to 9 points, with 3 being least lonely.

(9) EQ-5D visual analogue scale, from 0 to 100, higher indicating better health.

(10) Eysenck neuroticism scale.

stratified relative risks being above 1, the relative risk for the whole cohort are close to 1 for depression (1.07) and anxiety (0.98), suggesting little change between 2016 and 2022, or below 1 for alcohol use disorder (0.84), due to the gradient of fewer cases in older age groups.

## D: Consistency of lifetime phenotypes

Table 4 shows the difference in results from measures that report lifetime phenotypes in those who completed both MHQ1 and MHQ2: two symptom-based definitions of lifetime disorder (depression and bipolar), two lifetime behaviours (self-harm and cannabis use), and self-report of any clinician diagnosis. The criteria for positive status were the same in both questionnaires. Across these phenotypes, respondents seem more likely to meet the criteria in MHQ1 than in MHQ2, except for self-harm. Kappa values for the two symptom-based definitions are depression 0.53 and bipolar 0.30, with values being higher in the other categories (any clinician diagnosis 0.66, self-harm 0.67, cannabis 0.81). A potential cause of lack of agreement between the questionnaire waves are cases that started after completing MHQ1. Looking at depression, of those 5,947 who were positive in MHQ2 but not MHQ1, at least 4,445 (75%) had an onset before the MHQ1 wave, so could have been detected then. For self-harm 77%, bipolar 90% and cannabis 93% of those who were negative in MHQ1 could have been detected according to self-reported onset, therefore new-onset can explain only a small amount of discordant cases. S5 and S6 Tables in S1 File show the sex-stratified results, and they are similar; for example, self-harm reporting in women had a kappa of 0.68, and in men 0.65.

## Discussion

This paper introduced UK Biobank's second mental health questionnaire. The original MHQ was one of the largest mental health surveys ever reported. This second wave gives an even larger sample size, more detail, and coverage of broader aspects of mental health. Responses from 169,253 UK Biobank participants are available for the MHQ2, which brings to 215,252 the number who have at least one wave of the mental health questionnaire that can be analysed. We have

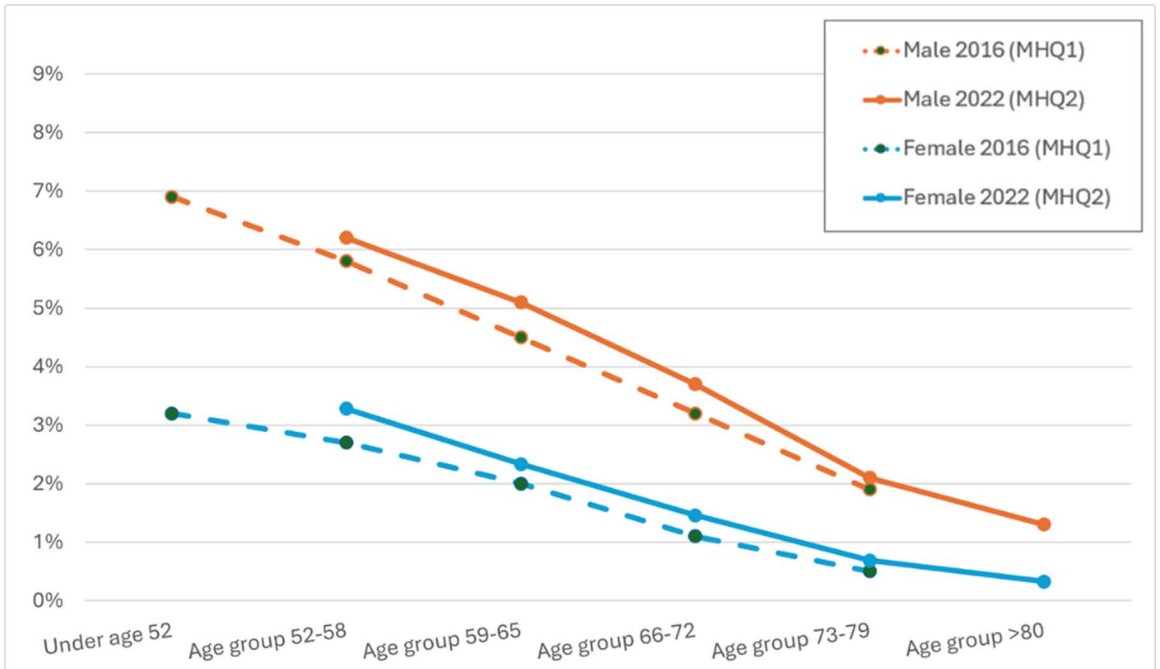

**Fig 4. Proportion in each age group (at completion of relevant questionnaire) with harmful alcohol use according to AUDIT by sex and wave, restricted to those who completed both MHQ1 ≈ 2016 and MHQ2 ≈ 2022 (n = 111,275).**

selected a few notable features to describe, which we hope will stimulate interest in this resource. These were: (A) who responded to the MHQ2; (B) the mental health phenotypes and social factors; (C) changes in the mental health of the cohort; and (D) the consistency of lifetime phenotypes.

Respondents: There was a substantial overlap of participants who completed the previous wave, and participant characteristics of the two waves were similar. As shown in the previous paper [17] the biases in completion of the mental health questionnaires resembled and amplified the biases in recruitment to UKB. The respondents to MHQ2 appeared less likely to be deprived or have a chronic illness than the overall UKB cohort, and 97% were from a White ethnic group.

Mental health and social factors: Depression was the most common lifetime disorder at 18% in the MHQ2 wave, although it was 24% in the MHQ1 wave. Next in this questionnaire was panic disorder (4.0%), eating disorder (2.8%) and bipolar type I (0.4%). The pattern of social and health characteristics appear to be different for people who met criteria for any of those four disorders, including suggestions of increased adverse child events, more difficult and deprived personal circumstances, and poorer health. This pattern seems more pronounced with rarer disorders. If those people in UK Biobank with a history of mental health disorder or disorders had worse self-rated health, this can be due to health being poorer both because of the mental disorders themselves or because they are at greater risk of developing physical illness or, conversely, due to them having a physical disorder that places them at higher risk of developing a mental disorder [9]. UK Biobank is a valuable data source for studying these issues, and repeated measures to give longitudinal data will help.

## Repeated measures

Current mental health: In the MHQ1 wave, both lifetime and current mental health disorders appeared to be rarer in old age. This could be an age effect (mental health gets better as people age) or a cohort effect (people born earlier have better mental health). Using repeated current mental disorder scales, the frequency of alcohol use disorder appeared to decrease

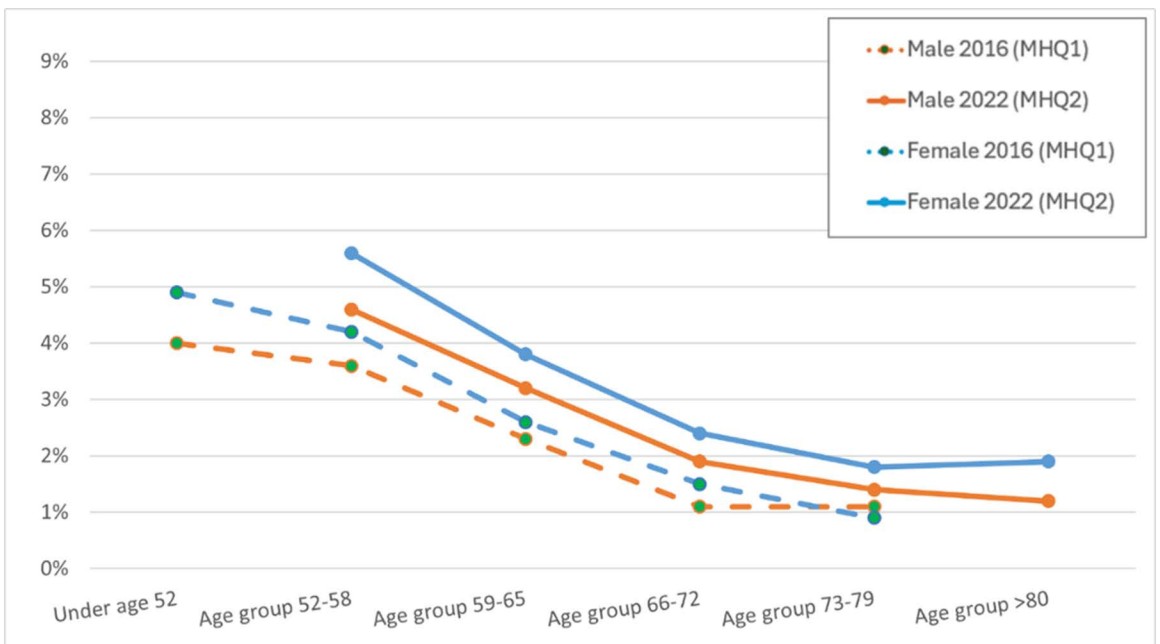

**Fig 5. Proportion in each age group (at completion of relevant questionnaire) with 'PHQ-9-derived current depression' outcome by sex and wave, restricted to those who completed both MHQ1 ≈ 2016 and MHQ2 ≈ 2022 (n = 111,275).**

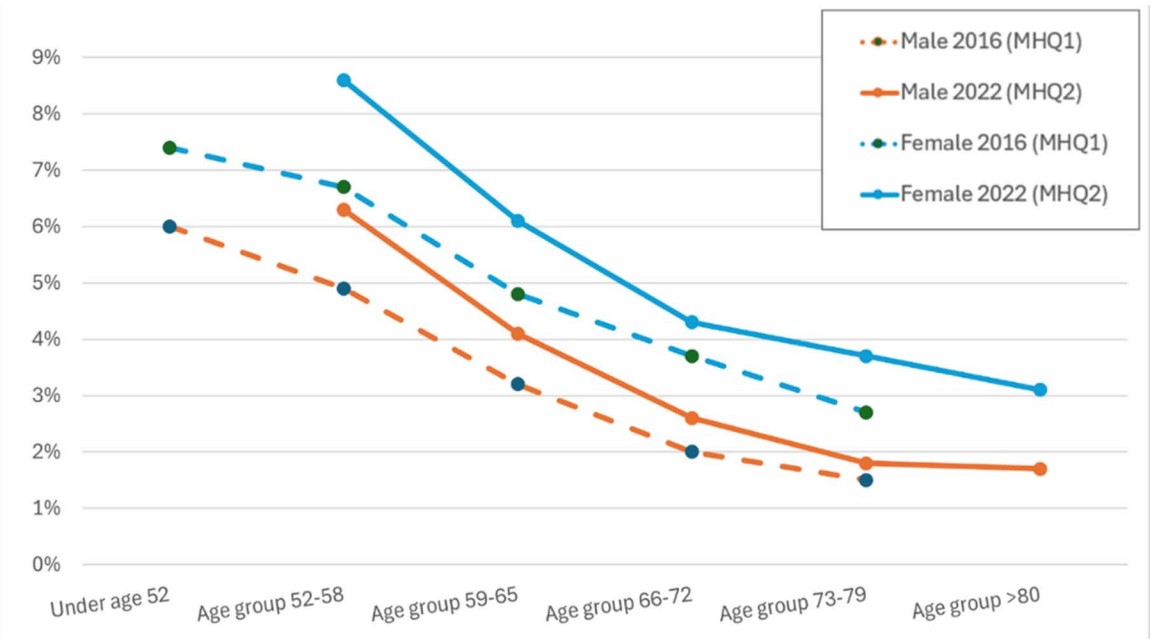

**Fig 6. Proportion in each age group (at completion of relevant questionnaire) with generalised anxiety disorder according to 'GAD-7-derived current anxiety' by sex and wave, restricted to those who completed both MHQ1 ≈ 2016 and MHQ2 ≈ 2022 (n = 111,275).**

**Table 4. Comparison of lifetime phenotype status in MHQ1 and MHQ2 in those participants who completed both questionnaires (n = 111,275). Percentage is proportion of those participants in each agreement category, except last column. Last column takes into account the date of onset in lack of agreement, percentage is proportion with onset prior to 2016 (approximate date of MHQ1).**

| Status MHQ1 | Positive | Positive | Negative | Negative | Agree (%) | Kappa | Negative |
|---|---|---|---|---|---|---|---|
| Status MHQ2 | Positive | Negative | Positive | Negative | | | Positive AND onset <2016* |
| Case criteria met | Both | 1 not 2 | 2 not 1 | Neither | | | % of 2 not 1 |
| Depression | 14,612/111,275 (13.1%) | 11,670/111,275 (10.5%) | 5,947/111,275 (5.3%) | 79,046/111,275 (71.0%) | 84% | 0.53 | 4,445/5,947 (74.7%) |
| Bipolar | 157/111,275 (0.1%) | 448/111,275 (0.4%) | 279/111,275 (0.3%) | 110,391/111,275 (99.2%) | 99% | 0.30 | 250/279 (89.6%) |
| Self-harm | 3,498/111,275 (3.1%) | 1,411/111,275 (1.3%) | 1,788/111,275 (1.6%) | 104,578/111,275 (94.0%) | 97% | 0.67 | 1,378/1,788 (77.1%) |
| Cannabis use | 21,630/111,275 (19.4%) | 4,015/111,275 (3.6%) | 3,393/111,275 (3.0%) | 82,237/111,275 (73.9%) | 93% | 0.81 | 3,139/3,393 (92.5%) |
| Any SR clinician diagnosis | 25,701/111,275 (23.1%) | 11,678/111,275 (10.5%) | 4,286/111,275 (3.9%) | 69,610/111,275 (62.6%) | 86% | 0.66 | NA |

\* date of onset (of depression, bipolar, self-harm or cannabis use) was reported in MHQ2 to be before MHQ1 wave, rather than onset that was definitely after MHQ1 wave or was unknown.

between MHQ1 and MHQ2 (six years), suggesting an age effect. For depression and anxiety, the frequency appeared to be nearly unchanged as the cohort aged, suggesting a cohort effect. Another possibility is that 2022 was an era of higher anxiety and depression due to the COVID pandemic and knock-on effects, for example, prolonged health service disruption, which may make it difficult for any age-related improvement in mental health to be demonstrated at this time. The short-term deviation of mental distress from the COVID pandemic has been well-documented (for example,[49]), but the long-term mental health outcomes of the period are still unknown [50], and may merit monitoring. There are already some measures that can be examined longitudinally, for instance, selected questions from the PHQ-9 were asked at baseline and in several enhancements (sleep, cognitive testing, imaging visits, MHQ1 and MHQ2) and the EQ-5D-5L (health-related quality of life) reported here from MHQ2 was also asked in the UKB pain questionnaire in 2019, and in the pain 2 questionnaire in 2024.

Lifetime phenotypes: We investigated the test-retest consistency for phenotypes that we would expect to be similar between waves: lifetime depression, bipolar affective disorder type I, self-harm, cannabis use and self-report of any clinician diagnosis. In all cases, there was high agreement on phenotype status (from 84% for depression to 99% for bipolar), although kappa (which corrects for agreement by chance) revealed that lifetime depression and lifetime bipolar statuses were somewhat inconsistent (0.53 for depression, 0.30 for bipolar), and this was not accounted for by new-onset. To meet criteria for depression and bipolar, several criteria needed to be met, frequently hinging on a dichotomous scoring of a Likert-like scale. For instance, for the question "In your worst ever episode… How much did these problems interfere with your life or activities? – A lot/ Somewhat/ A little/ Not at all", an answer of "A lot" or "Somewhat" would support a depression status, but "A little" would rule it out. We can speculate that a person's appraisal could change from "Somewhat" to "A little" over time, perhaps to a more positive interpretation as they age, or that the worst episode they were thinking about in 2016 was different in this respect from the one they were thinking about in 2022. Therefore, a change in status according from criteria may occur with only modest changes in reporting. To meet the criteria for bipolar, participants had to meet criteria for both depression and mania. The definition for ever self-harm and cannabis consisted of a few yes/no questions. For yes/no questions, and when talking about behaviours rather than feelings, there would be less reappraisal. It is not clear why the recall of self-report clinician diagnosis might be inconsistent, perhaps it hinges on recall of whether a disorder was diagnosed, rather

than merely discussed or mentioned. These inconsistencies may complicate the interpretation of some of these outcomes, but it is possible they reflect the underlying uncertain boundaries between health and disorder [51,52].

### Strengths and limitations

UK Biobank is a valuable resource for investigating mental health, but it also presents challenges. The cohort consists of a convenience sample, and the non-representativeness observed at baseline assessments is exacerbated in the voluntary enhancements (such as the MHQs). The multiple enhancements with different samples and timings can complicate planning and interpreting analyses. We can speculate that people with poorer health, including mental health, would be less likely to start a UK Biobank questionnaire, or may have not been able to complete (6,000 people or 3% of those that started did not complete). Because of non-representativeness, researchers should not make population inferences based solely on UK Biobank data. We have not tested what effect the non-representativeness of the sample has had on our ascertainment of mental disorders and the pattern with social factors.

We have designed the UKB MHQ1 and MHQ2 with researchers familiar with large cohorts and clinical academics – leading to the use of widely accepted measures and clinically relevant outcomes. The completion of the questionnaire was also high once started, assisted by reminder emails. However, the MHQs are reliant on self-report, frequently of an episode of psychopathology that may have been many years ago. This may result in some inconsistencies, such as seen in the test-retest statistics – which we suggest is due to inconsistent recall and indistinct boundaries in mental disorders, which have no diagnostic tests.

Our approach to developing this paper and facilitating further research using MHQ2 has incorporated quality assurance processes when developing algorithms and code, which we are sharing with the wider research community. We enhance transparency by sharing resources, however, errors are still inevitable in such projects [53] so we encourage caution when using these resources, and feedback from others if they suspect any problems. This paper is just the beginning, which we hope will encourage other researchers to investigate more deeply. Further advice for people considering using UKB for mental health research is available in a recent review paper [20].

### Conclusion

The mental health questionnaires enhance UKB with large amounts of data on common mental disorders (particularly depression), and also on disorders that have not received much research in an older population, such as panic disorder and eating disorders, and update social factors a decade after baseline. Importantly, the UKB now has included transdiagnostic symptoms (self-harm, body weight/shape preoccupation in MHQ2, psychotic experiences in MHQ1) and psychological constructs (resilience, loneliness and neuroticism) that are infrequently or never available from clinical data. Repeated measures for items such as depressive symptoms and health-related quality of life, along with linkage to routine health data for follow-up, allow longitudinal and causal analyses. UKB offers a large sample size, extensive genetic information, biomarkers and imaging data, alongside detailed mental health data, which we hope will be used to help us understand more about mental health, and how we might prevent and treat poor mental health in the future.

### Supporting information

**S1 File. S1 Fig, S1-6 Tables.**
(PDF)

### Author contributions

**Conceptualization:** Katrina A S Davis, Jonathan R I Coleman, Mark Adams, Gerome Breen, Helena L Davies, Alexandru Dregan, Thalia C Eley, Elaine Fox, Jo Holliday, Christopher Hübel, Ann John, William Lee, Rose McCabe, Andrew M McIntosh, Sian Oram, Marcus Richards, Fenella Starkey, Laura M Thornton, Matthew Hotopf.

**Data curation:** Jonathan R I Coleman, Mark Adams, Helena L Davies, Kelly Davies, Christopher Hübel, Danyang Li, Jared Maina, Megan Skelton, Abigail R ter Kuile, Rujia Wang, Zhaoying Yu, Johan Zvrskovec.

**Formal analysis:** Katrina A S Davis, Jonathan R I Coleman, Kelly Davies.

**Methodology:** Jonathan R I Coleman, Fenella Starkey, Matthew Hotopf.

**Project administration:** Katrina A S Davis, Jonathan R I Coleman.

**Resources:** Matthew Hotopf.

**Supervision:** Matthew Hotopf.

**Validation:** Christopher Hübel.

**Visualization:** Aliyah S Kassam.

**Writing – original draft:** Katrina A S Davis.

**Writing – review & editing:** Katrina A S Davis, Jonathan R I Coleman, Mark Adams, Gerome Breen, Na Cai, Helena L Davies, Kelly Davies, Alexandru Dregan, Thalia C Eley, Elaine Fox, Jo Holliday, Christopher Hübel, Ann John, Aliyah S Kassam, Matthew J Kempton, William Lee, Danyang Li, Jared Maina, Rose McCabe, Andrew M McIntosh, Sian Oram, Marcus Richards, Megan Skelton, Fenella Starkey, Abigail R ter Kuile, Laura M Thornton, Rujia Wang, Zhaoying Yu, Johan Zvrskovec, Matthew Hotopf.

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
