## [Decision Letter · Decision Letter 0]

19 Feb 2025

PONE-D-24-54764The UK Biobank Mental Health Enhancement 2022: Methods and ResultsPLOS ONE

Dear Dr. Davis,

Thank you for submitting your manuscript to PLOS ONE. After careful consideration, we feel that it has merit but does not fully meet PLOS ONE’s publication criteria as it currently stands. Therefore, we invite you to submit a revised version of the manuscript that addresses the points raised during the review process.

We look forward to receiving your revised manuscript.

Kind regards,

Sandar Tin Tin

Academic Editor

PLOS ONE

Journal Requirements:

“I have read the journal's policy and the authors of this manuscript have the following competing interests: AJ received a fee to talk at the centenary of the Scottish Action for Mental Health (SAMH). The other authors declare no conflict of interest.”

5. We note that you have indicated that there are restrictions to data sharing for this study. PLOS only allows data to be available upon request if there are legal or ethical restrictions on sharing data publicly. For more information on unacceptable data access restrictions, please see http://journals.plos.org/plosone/s/data-availability#loc-unacceptable-data-access-restrictions.

6. Please remove your figures from within your manuscript file, leaving only the individual TIFF/EPS image files, uploaded separately. These will be automatically included in the reviewers’ PDF.

Reviewers' comments:

Reviewer's Responses to Questions

**Comments to the Author**

1. Is the manuscript technically sound, and do the data support the conclusions?

Reviewer #1: Yes

Reviewer #2: Yes

2. Has the statistical analysis been performed appropriately and rigorously? 

Reviewer #1: Yes

Reviewer #2: Yes

3. Have the authors made all data underlying the findings in their manuscript fully available?

Reviewer #1: No

Reviewer #2: Yes

4. Is the manuscript presented in an intelligible fashion and written in standard English?

Reviewer #1: Yes

Reviewer #2: Yes

5. Review Comments to the Author

Reviewer #1: Summary of the work: This study reports findings from the UK Biobank’s second mental health questionnaire wave. First, they explain the question changes from MHQ1 to MHQ2. Then they evaluate many characteristics of MHQ2, including comparisons between MHQ1 and MHQ2. This work not only provides a useful understanding of the mental health progression of the UKB participants, but it also provides a useful test-retest evaluation of the lifetime prevalence measures of the UKB.

Overall thoughts: This study is an exciting contribution to the field that will allow many researchers to expand their investigation of mental health in the UK Biobank. Their explanation of the differences in questions, participant overlap, and demographics between MHQ1 and MHQ2 are clear and valuable to the field. The figures are nice and effectively convey the information they intend to. Their analytical evaluation of MHQ2 however could be improved by further methodological explanation and potentially increased statistical rigor. Some analyses are performed with minimal explanation and some comparison results have no stated statistical analyses where the authors may wish to include them. With that said, the comparisons they provided are useful and important for researchers of the field to see. Therefore, I excitedly endorse its acceptance at PLOS ONE after minor revisions.

Minor comments:

1. Methodological clarity would be improved if the sections in the Methods that explain the analyses/comparisons of results were more clearly linked. For example using 1, 2, 3/A, B, C/or identical headers to link the method and its associated result.

2. Many methods, particularly analyses, do not seem to be seem to be explained in detail anywhere in the paper or supplement. This not only often leaves the reader with confusion about what analysis was done but also how to evaluate the results. Relatedly, many tables seem to include statistical quantifications that are not explained in the paper or table legend.

a. For example, please elaborate on how relative risk for depression and anxiety, stated in line 376, were

calculated. The authors state that the relative risk for these disorders are approximately the same, was this

determined using statistical analysis?

b. For example, details of the evaluation of lifetime diagnosis between MHQ1 and MHQ2, and its associated Table

4, are unclear. Particularly, the line “after taking into account onsets after MHQ1 (including unknown onsets),

between 75% to 93% of the test-retest variation in direction ‘Met 2 not 1’ and all in direction ‘Met 1 not 2’

remains unexplained” should be improved for clarity. Please explain how unknown onsets are incorporated (I assume the authors included any unknown onset into ‘onsets after MHQ1’ which seems an odd choice because it is not clear why these unknown onsets would be more likely to occur after MHQ1 as opposed to before). Additionally, how is test-retest variation determined and what does 75%-93% variation mean for this finding? What indicates that this variation is unexplained? Table 4, which relates to this result, could also be improved by further elaboration. What does the “Agree (%)” column refer to? The column “New onset (% Met 2 not 1)” should also explicitly state in the legend that this is the percent of individuals in the Met 2 not 1 group that reported new onset since MHQ1. Please also explain how new onset was determined.

3. The results include many comparisons between quantitative measures, but do not state a statistical difference. For example, starting on line 343, “those with lifetime eating disorders are more likely to have a degree qualification (eating disorder 53%; depression 46%; no mental disorder 44%)” compares the degree qualification of these 3 groups but not does indicate if these groups are statistically different. I recognize that this work is intended more as a descriptor for the new MHQ2, but I would refrain from drawing conclusions about these comparisons without statistical analysis. I recommend either including this statistical analysis or dampening claims about such comparisons throughout the paper.

4. For the methodological explanation starting at line 256, “Age at MHQ2…”, why was age of participants who did not complete MHQ2 included in the average age calculated for MHQ2? I assume that this age calculation is used in the demographics comparison table, which seems problematic as this does not accurately reflect the age of the MHQ2 participants to my mind. Though perhaps I misunderstand the explanation of this methodology, in which case I recommend improving the clarity of writing for this section.

5. I recognized that criteria for lifetime status is unchanged from MHQ1, but it may be of value to restate these criteria in this work.

6. I assume “dx” in Table 3 is short for diagnosis. Please include this abbreviation in the table legend.

Reviewer #2: This manuscript describes the procedure and content of the UK Biobank (UKB) MHQ2 questionnaire, provides a general cohort description, and presents overall comparisons between the two mental health data collections in UKB. The authors report a significant enhancement to the UKB resource by conducting a second mental health data collection in 169,253 individuals, bringing the total number of UKB participants with mental health data from at least one time point to 215,252, and those with data from two time points to 111,275. This represents a major advancement for the UKB mental health phenotypes and is an important contribution to the research field.

The manuscript clearly outlines the rationale and procedural decisions made during the design phase. The authors describe the characteristics of participants across the two waves and the broader UKB cohort, as well as across symptom-based criteria for lifetime disorders. They also examine changes in mental health using ‘current’ assessments and assess the consistency of measures using ‘lifetime’ assessments.

Comments and suggestions

1. It appears that the authors defined participants based on completion of all questionnaire modules. Given the difference between the number of individuals who started (175,266) and those who completed (169,252), 6,014 individuals did not finish the questionnaire. Inability to complete the full questionnaire may be linked to more severe mental health conditions, such as difficulties with concentration, fatigue, or other relevant health issues. The responses provided by these individuals before discontinuation could still be valuable for analyzing MHQ2 data. It would be helpful if the authors could clarify their decision to exclude these individuals in this context.

2. If data on completion times are available, it would be informative to include basic descriptive statistics on the time required to complete the full questionnaire. This would help estimate the time investment expected from participants.

3. Is it possible to determine whether the worst episodes described in detail in MHQ1 and MHQ2 refers to the same episode or two different episodes? This distinction would be useful both for assessing changes in symptoms across different episodes and for evaluating recall inconsistency for symptoms of the same episode. As the questionnaire does not seem to ask about the age at which the worst episode occurred, making this distinction may be challenging. Clarification on this point would be valuable.

4. It is unclear why psychotic experiences were omitted from MHQ2. While balancing questionnaire length and topic coverage is important, multiple assessments of psychotic symptoms could be highly relevant for investigating their links to, for example, cannabis use, medication use, or changes over the second half of the lifespan. Related to this, the wording in lines 502–504 of the Conclusions section may unintentionally suggest that psychotic experiences were included in this data collection. Rewording this sentence for clarity is recommended.

5. There appears to be a minor inconsistency in the reported sample size. The abstract and Tables 2 and 3 state that the number of respondents was 169,253, while the Results section (line 276) and Figure 2 state 169,252. It would be helpful to double-check and ensure consistency across the manuscript.

6. PLOS authors have the option to publish the peer review history of their article (what does this mean? ). If published, this will include your full peer review and any attached files.

**Do you want your identity to be public for this peer review?** For information about this choice, including consent withdrawal, please see our Privacy Policy .

Reviewer #1: No

Reviewer #2: No

---

## [Author Response · Author response to Decision Letter 1]

1 Apr 2025

Reviewer #1: Summary of the work: This study reports findings from the UK Biobank’s second mental health questionnaire wave. First, they explain the question changes from MHQ1 to MHQ2. Then they evaluate many characteristics of MHQ2, including comparisons between MHQ1 and MHQ2. This work not only provides a useful understanding of the mental health progression of the UKB participants, but it also provides a useful test-retest evaluation of the lifetime prevalence measures of the UKB.

Overall thoughts: This study is an exciting contribution to the field that will allow many researchers to expand their investigation of mental health in the UK Biobank. Their explanation of the differences in questions, participant overlap, and demographics between MHQ1 and MHQ2 are clear and valuable to the field. The figures are nice and effectively convey the information they intend to. Their analytical evaluation of MHQ2 however could be improved by further methodological explanation and potentially increased statistical rigor. Some analyses are performed with minimal explanation and some comparison results have no stated statistical analyses where the authors may wish to include them. With that said, the comparisons they provided are useful and important for researchers of the field to see. Therefore, I excitedly endorse its acceptance at PLOS ONE after minor revisions.

Minor comments:

1. Methodological clarity would be improved if the sections in the Methods that explain the analyses/comparisons of results were more clearly linked. For example using 1, 2, 3/A, B, C/or identical headers to link the method and its associated result.

Response: Thank you for this suggestion. We have now lettered our aims, and used these to structure the analysis section and results. We aim to report (A) who responded to the MHQ1 and MHQ2, (B) the mental health phenotypes and social factors ascertained in MHQ2, (C) changes in the current mental health of the cohort, and (D) the consistency of lifetime phenotypes.

2. Many methods, particularly analyses, do not seem to be seem to be explained in detail anywhere in the paper or supplement. This not only often leaves the reader with confusion about what analysis was done but also how to evaluate the results. Relatedly, many tables seem to include statistical quantifications that are not explained in the paper or table legend.

Response: We have rewritten the analysis section to be sure that it describes those analyses that follow.

a. For example, please elaborate on how relative risk for depression and anxiety, stated in line 376, were calculated.

Response: Thank you for highlighting this. We now state in the analysis section that:

“Changes in Current Mental Health: We describe and visualise the proportion of respondents who were positive for specified current mental disorders in 2022 and in 2016 in those participants who answered in both waves… Relative risk for 2022 compared to 2016 was calculated by using ‘proportion positive’ as the risk of disorder in each cohort and then risk in 2022 by that in 2016. This is done in age-and-sex stratified groups such that those aged, for example, 73-79 years in 2016 are compared with those aged 73-79 in 2022 for the four age groups where possible and also for the cohort overall.”

The authors state that the relative risk for these disorders are approximately the same, was this determined using statistical analysis?

Thank you, the answer is no. We agree that this may have sounded as though we were making a formal comparison where we had not intended to do so. We have clarified in the methods: “Significance testing on large cohorts may identify many small differences between comparators that have very little effect (35), therefore we have emphasised the magnitude of difference rather than calculating p-values.”

We have also rephrased this section for greater clarity:

“Supplementary Table 4 quantifies this a little more by reporting the relative risk of each current mental disorder in 2022 versus 2016 in each sex and age-group stratification and overall. A relative risk of 1 would suggest no change of risk in the group for the current disorder in 2022 compared to 2016 wave, with values above 1 suggesting a higher risk in 2022. The relative risk for every sex and age-group stratified group was above 1, between 1.07 and 2.00, with the latter representing a doubling of the presence of current depression in women aged 73 to 79 in 2022 compared to 2016. Despite age-group stratified relative risks being above 1, the relative risk for the whole cohort are close to 1 for depression (1.07) and anxiety (0.98), suggesting little change between 2016 and 2022, or below 1 for alcohol use disorder (0.84), due to the gradient of fewer cases in older age groups.”

b. For example, details of the evaluation of lifetime diagnosis between MHQ1 and MHQ2, and its associated Table 4, are unclear. Particularly, the line “after taking into account onsets after MHQ1 (including unknown onsets), between 75% to 93% of the test-retest variation in direction ‘Met 2 not 1’ and all in direction ‘Met 1 not 2’ remains unexplained” should be improved for clarity.

Please explain how unknown onsets are incorporated (I assume the authors included any unknown onset into ‘onsets after MHQ1’ which seems an odd choice because it is not clear why these unknown onsets would be more likely to occur after MHQ1 as opposed to before). Additionally, how is test-retest variation determined and what does 75%-93% variation mean for this finding? What indicates that this variation is unexplained? Table 4, which relates to this result, could also be improved by further elaboration. What does the “Agree (%)” column refer to? The column “New onset (% Met 2 not 1)” should also explicitly state in the legend that this is the percent of individuals in the Met 2 not 1 group that reported new onset since MHQ1. Please also explain how new onset was determined.

Response: We agree that our description of onset timing and discrepancy of findings was not clear. Regarding new-onset, we knew that readers would want to know how much of the discrepancy between MHQ1 and MHQ2 were due to this, so we extracted data to distinguish between those that should have been picked up by MHQ1 from those we were less sure or knew could not have been picked up. Although we cannot know the onset when respondents marked ’don’t know’, we included them along with documented recent onset in order to give the proportion that definitely could have been detected at MHQ1 if a true disorder. For clarity, we now only report those figures: of those that were negative in MHQ1 and positive in MHQ2, the proportion that, from their date of onset, also had the disorder at the time of the MHQ1. This meant losing a small section in the discussion that was about the magnitude of new-onset cases that, on reflection, was speculative and not part of our aims.

We have made changes to the analysis description in the methods, results section and table, which we hope explain more of our intentions and findings.

In methods we have written:

“D: Consistency of lifetime phenotypes…. Percentage agreement for each status was calculated as status agreeing in 2016 and 2022 (both positive or both negative) as a proportion of the cohort, and Cohen’s kappa for each status, with kappa being agreement corrected for agreement by chance(38). Where a respondent was negative in 2016 but positive in 2022, we also took into account the date that the respondent gave for onset in MHQ2 (except for clinician diagnosis, where this was not asked). This helps separate these respondents into those whose status could have been picked up in the MHQ1 wave (date of onset from MHQ2 was before 2016), from those who might not have done (either date of onset after start of 2016 or answered ‘don’t know’ to date of onset). We decided to include unknown with the later onsets, despite no information to suggest these are more likely to be late onset, to explain the greatest amount of test-retest variability.”

In results we have written:

“A potential cause of lack of agreement between the questionnaire waves are new-onset cases that started after completing MHQ1. Of those 5,947 who were positive in MHQ2 but not MHQ1 for depression, 4,445 (75%) reported an onset before the MHQ1 wave, so could have been detected then. For self-harm 77%, bipolar 90% and cannabis 93% of those who were negative in MHQ1 could have been detected according to self-reported onset, therefore new-onset explains only a small amount of discordant cases.”

For the table, we have added some additional label rows, so that rather than a column being labelled “Met 1 not 2” it is labelled as ‘Positive’ in the MHQ1 status row, ‘Negative’ in the MHQ2 row, and so on, as we think this will be more intuitive. We have also put the denominator into the cells for clarity. The legend has been changed to: “Comparison of lifetime phenotype status in MHQ1 and MHQ2 in those participants who completed both questionnaires (n=111,275). Percentage is proportion of those participants in each agreement category, except last column. Last column takes into account the date of onset in lack of agreement, percentage is proportion with onset prior to 2016 (approximate date of MHQ1).” And with a footnote “*date of onset (of depression, bipolar, self-harm or cannabis use) was reported in MHQ2 to be before MHQ1 wave, rather than onset that was definitely after MHQ1 wave or was unknown.”

3. The results include many comparisons between quantitative measures, but do not state a statistical difference. For example, starting on line 343, “those with lifetime eating disorders are more likely to have a degree qualification (eating disorder 53%; depression 46%; no mental disorder 44%)” compares the degree qualification of these 3 groups but not does indicate if these groups are statistically different. I recognize that this work is intended more as a descriptor for the new MHQ2, but I would refrain from drawing conclusions about these comparisons without statistical analysis. I recommend either including this statistical analysis or dampening claims about such comparisons throughout the paper.

Response: Thank you. It is true that we wish to be mainly descriptive. Another reason for being reluctant to include significance testing is that the large sample size means that many comparisons will be statistically significant. Therefore, we have stated in the analysis plan: “Significance testing on large cohorts may identify many small differences between comparators that have very little effect (35), therefore we have emphasised the magnitude of difference rather than calculating p-values.” We have made minor changes to the results sections to avoid implying we have made formal comparisons. For the result mentioned above, we have changed this to read:

“The group of respondents that met the criteria for eating disorders and panic disorder appear to have similar characteristics to those respondents with depression and bipolar, although it was remarkable that over half of those with lifetime eating disorders had a degree qualification (53%).”

4. For the methodological explanation starting at line 256, “Age at MHQ2…”, why was age of participants who did not complete MHQ2 included in the average age calculated for MHQ2? I assume that this age calculation is used in the demographics comparison table, which seems problematic as this does not accurately reflect the age of the MHQ2 participants to my mind. Though perhaps I misunderstand the explanation of this methodology, in which case I recommend improving the clarity of writing for this section.

Response: Thank you, we hope we have now provided more clarity on this. For the group that answered the MHQ2, their age is taken as when they completed the questionnaire, and this is what is presented in table 2 in the MHQ2 column. Our dilemma was that most in the full cohort did not complete the questionnaire, so we had to find an equivalent point in order to compare the MHQ2 respondents with the base cohort, which we decided was the median data of completion of the MHQ2. We then applied this to everyone (regardless of whether had completed MHQ2 or not), and that is what is presented in table 2 in the UK Biobank cohort column.

After revisions, in the methods section we say:

“The age variable was handled differently for the three groups. Age for the MHQ2 wave was their age on completion of the MHQ2, and for the MHQ1 wave age at completion of MHQ1. For the UKB cohort as a whole, age represents the age they would have been at the median date of MHQ2 completion, including those who died before this date”

In the footnote to table 2 we say:

“Age in the MHQ2 wave is at date of completion of MHQ2. Age in the MHQ1 wave is at approximate date of completion of MHQ1. Age for full cohort is the age a participant would have been on the median date of MHQ2 completion.”

5. I recognized that criteria for lifetime status is unchanged from MHQ1, but it may be of value to restate these criteria in this work.

Response: Thank you for the suggestion. We have added :

“The criteria for positive status were the same in both questionnaires.”

6. I assume “dx” in Table 3 is short for diagnosis. Please include this abbreviation in the table legend.

Response: Thanks, added.

Reviewer #2: This manuscript describes the procedure and content of the UK Biobank (UKB) MHQ2 questionnaire, provides a general cohort description, and presents overall comparisons between the two mental health data collections in UKB. The authors report a significant enhancement to the UKB resource by conducting a second mental health data collection in 169,253 individuals, bringing the total number of UKB participants with mental health data from at least one time point to 215,252, and those with data from two time points to 111,275. This represents a major advancement for the UKB mental health phenotypes and is an important contribution to the research field.

The manuscript clearly outlines the rationale and procedural decisions made during the design phase. The authors describe the characteristics of participants across the two waves and the broader UKB cohort, as well as across symptom-based criteria for lifetime disorders. They also examine changes in mental health using ‘current’ assessments and assess the consistency of measures using ‘lifetime’ assessments.

Comments and suggestions

1. It appears that the authors defined participants based on completion of all questionnaire modules. Given the difference between the number of individuals who started (175,266) and those who completed (169,252), 6,014 individuals did not finish the questionnaire. Inability to complete the full questionnaire may be linked to more severe mental health conditions, such as difficulties with concentration, fatigue, or other relevant health issues. The responses provided by these individuals before discontinuation could still be valuable for analyzing MHQ2 data. It would be helpful if the authors could clarify their decision to exclude these individuals in this context.

Response: Thank you for this suggestion. We agree there is good reason to think that there may be a systematic bias against the very people who would be cases had they completed the questionnaire. However, in the absence of being able to test this, we are not able to comment on this. It may be helpful to know that the data is available on these people for all of the sections that they managed to fill out, but for the sake of simplicity in this report (different ns for different disorders etc.), we have omitted them.

In the strengths and limitations section we have written: “We can speculate that people with poorer health, including mental health, would be less likely to start a UK Biobank questionnaire, or may have not been able to complete (as around 6,000 people or 3% of those that started).”

2. If data on completion times are available, it would be informative to include basic descriptive statistics on the time required to complete the full questionnaire. This would help estimate the time investment expected from participants.

Response: We see that this would be useful. Unfortunately, we do not have this in any form that would be robust enough to quote. It is po

---

## [Decision Letter · Decision Letter 1]

22 Apr 2025

The UK Biobank mental health enhancement 2022: methods and results

PONE-D-24-54764R1

Dear Dr. Davis,

We’re pleased to inform you that your manuscript has been judged scientifically suitable for publication and will be formally accepted for publication once it meets all outstanding technical requirements.

Kind regards,

Sandar Tin Tin

Academic Editor

PLOS ONE

Additional Editor Comments (optional):

Reviewers' comments:

Reviewer's Responses to Questions

**Comments to the Author**

1. If the authors have adequately addressed your comments raised in a previous round of review and you feel that this manuscript is now acceptable for publication, you may indicate that here to bypass the “Comments to the Author” section, enter your conflict of interest statement in the “Confidential to Editor” section, and submit your "Accept" recommendation.

Reviewer #1: All comments have been addressed

2. Is the manuscript technically sound, and do the data support the conclusions?

Reviewer #1: Yes

3. Has the statistical analysis been performed appropriately and rigorously? 

Reviewer #1: Yes

4. Have the authors made all data underlying the findings in their manuscript fully available?

Reviewer #1: Yes

5. Is the manuscript presented in an intelligible fashion and written in standard English?

Reviewer #1: Yes

6. Review Comments to the Author

Reviewer #1: (No Response)

7. PLOS authors have the option to publish the peer review history of their article (what does this mean? ). If published, this will include your full peer review and any attached files.

**Do you want your identity to be public for this peer review?** For information about this choice, including consent withdrawal, please see our Privacy Policy .

Reviewer #1: No

---

## [Editor Report · Acceptance letter]

PONE-D-24-54764R1

PLOS ONE

Dear Dr. Davis,

I'm pleased to inform you that your manuscript has been deemed suitable for publication in PLOS ONE. Congratulations! Your manuscript is now being handed over to our production team.

Kind regards,

on behalf of

Dr. Sandar Tin Tin

Academic Editor

PLOS ONE